# Multi-Party Quantum Secret Sharing Based on GHZ State

**DOI:** 10.3390/e24101433

**Published:** 2022-10-08

**Authors:** Zhihui Li, Xue Jiang, Lu Liu

**Affiliations:** School of Mathematics and Statistics, Shaanxi Normal University, Xi’an 710119, China

**Keywords:** quantum secret sharing, GHZ state, local measurement, information efficiency

## Abstract

In this paper, we propose an efficient multi-party quantum secret sharing scheme based on GHZ entangled state. The participants in this scheme are divided into two groups, and share secrets as a group. There is no need to exchange any measurement information between the two groups, reducing the security problems caused by the communication process. Each participant holds one particle from each GHZ state; it can be found that the particles of each GHZ state are related after measuring them, and the eavesdropping detection can detect external attacks based on this characteristic. Furthermore, since the participants within the two groups encode the measured particles, they can recover the same secrets. Security analysis shows that the protocol can resist the intercept-and-resend attack and entanglement measurement attack, and the simulation results show that the probability of an external attacker being detected is proportional to the amount of information he can obtain. Compared with the existing protocols, this proposed protocol is more secure, has less quantum resources and is more practical.

## 1. Introduction

In recent years, with the maturity of quantum theory, quantum communication has developed rapidly. At present, the main branches of quantum communication include quantum key distribution [1,2,3,4], quantum secret sharing (QSS) [5,6,7,8], quantum digital signature [9,10], quantum authentication [11,12], quantum secure direct communication [13,14], etc. As an important branch, quantum secret sharing has always been the focus of attention. We usually describe a classic secret sharing as follows: suppose Alice has a secret task that needs to be completed in another place, but she cannot arrive in time. Fortunately, she has two assistants Bob and Charlie at the destination, but Alice does not trust either of them to perform the task alone. So Alice divides her task message into two parts and sends them to Bob and Charlie, respectively. Only the two of them can unite to recover Alice’s mission message, and no one can get Alice’s mission message alone. In this way, we have achieved the goal of managing secrets by multiple people and dispersing risks. However, with the development of science and technology, people have higher and higher requirements for security in the communication process. Classical secret sharing can no longer resist the eavesdropping attacks of modern technology. Since 1999, when Hillery et al. [5] proposed the first quantum secret sharing scheme (HBB99 protocol) based on the Greenberger–Horne–Zeilinger (GHZ) entangled state, quantum secret sharing has developed rapidly. Many domestic and foreign scholars have used various approaches to construct secret sharing systems, such as quantum error correction code [15], continuously variable cluster state [16], dense coding [17], Grover algorithm [18], etc.

According to the different methods based on physical resources and carrying information, quantum secret sharing schemes can be roughly divided into three types: the first type is the QSS scheme based on product states [19,20,21,22]; the second type is the QSS scheme based on single photon [23,24,25,26,27,28]. For example, in 2018, Bai et al. [26] proposed a new and efficient quantum secret sharing protocol using a d-level single particle, which can realize a general access structure through the idea of cascade. In 2020, Sutradhar et al. [27] proposed a secure *d*-level QSS protocol to share secrets, which could be reconstructed by *t* participants without trusted participants. Compared with most QSS protocols, this protocol was more secure, flexible and practical. In 2021, Chou et al. [28] proposed a novel method to share quantum information and established a (w,ω,n) multi-party weighted threshold quantum secret sharing scheme based on the Chinese Remainder Theorem (CRT) and phase shift operation. The third category QSS scheme is based on entanglement [29,30,31,32,33,34]. Among them, in the entangled state-based QSS scheme, researchers have done a lot of work on the efficiency and security of the protocol. In terms of efficiency, Tong Xin et al. [29] proposed a quantum secret sharing scheme based on GHZ state entanglement exchange in 2007. In this scheme, two GHZ state entanglement exchanges could share 2bit classical messages, which doubled the efficiency compared with the HBB protocol and the KKI protocol [30]. In 2008, Deng et al. [31] improved the KKI protocol and proposed an efficient large-capacity key encoding scheme with the efficiency increased to 50%. In 2014, Liao et al. [32] proposed a three-way dynamic quantum secret sharing scheme based on the GHZ state, which achieved the highest efficiency compared with the existing dynamic quantum secret sharing schemes. In 2019, Song Yun [33] proposed a quantum secret sharing scheme based on the local measurement of three particle GHZ states. When the number of detected GHZ quantum states is equal to the number of information GHZ quantum states, the efficiency of this scheme can reach 50%, and it does not require unitary operation. Song’s scheme is relatively economical in quantum resources, but limited in the number of participants.

In this paper, we propose a quantum secret sharing scheme based on the n(n⩾3) particle GHZ state, realize the quantum secret sharing among multiple parties, and the secrets are shared between the two groups. In our scheme, each participant holds a particle sequence of the GHZ state, and the measurement results of the same GHZ state can be found to be related by measurement. Therefore, we use this correlation to detect whether there is external attack in eavesdropping. In the recovery phase, the two groups do not need to exchange any information, and the shared secret can be obtained through the internal measurement and coding of each group, which reduces the external eavesdropping caused by the communication process. In addition, there is no unitary transformation in the transmission of this scheme.

This paper is structured as follows. In Section 2, we introduce the system definition. In Section 3, the protocol of the proposed scheme is given. In Section 4, we consider the intercept-and-resend attack and the entanglement measurement attack, and analyze the security simulation of this scheme. In Section 5, we analyze the efficiency and compare the performance of the proposed scheme. In Section 6, the quantum secret sharing schemes based on four-particle GHZ entangled states are listed. Section 7 gives our conclusion.

## 2. System Definition

### 2.1. System Model

In this paper, we construct a QSS scheme, which includes *n* participants $P_{1},P_{2},\ldots ,P_{n}$, and the *n* participants are divided into two groups $P_{A}$ and $P_{B}$ with $P_{A}=\left\{P_{A_{1}},P_{A_{2}},\ldots ,P_{A_{p}}\right\}$ and $P_{B}=\left\{P_{B_{1}},P_{B_{2}},\ldots ,P_{B_{q}}\right\}$, where n=p+q, *q* is an even number. Participant $P_{A_{1}}$ is the group leader of group $P_{A}$, and participant $P_{B_{1}}$ is the group leader of group $P_{B}$. This scheme needs to use two GHZ entangled states, respectively, i.e., ${|GHZ_{0}\rangle }_{1,\ldots ,n}=\frac{1}{\sqrt{2}}(|00\ldots 0\rangle +\left|11\ldots 1\rangle \right)$, ${|GHZ_{1}\rangle }_{1,\ldots ,n}=\frac{1}{\sqrt{2}}(|00\ldots 0\rangle -\left|11\ldots 1\rangle \right)$, where 1,…,n represents *n* particles.

### 2.2. Threat Model

In the attack model, we assume that participants are all honest and fully comply with the rules of this protocol. The external attacker Eve intercepts information through the channel. For the external attacker Eve, we use $I_{Eve}$ to represent the amount of information Eve can acquire, and *f* to represent the probability that the cheater is detected. If Eve obtains more than half of the information $I_{Eve}$, and the probability of being detected is greater than $\frac{1}{2}$ at the same time, the number of detected particles L≥3.3029 (see Section 4.3 for the proof process). In other words, as the number of detected particles *L* increases with L≥4, the probability of Eve being detected increases. Therefore, in the example of Section 6, we choose the number of detected particles L=4.

In addition, there is no need to communicate with each other through any classical channel or quantum channel for each group of participants in the same laboratory, and there is no possibility of external eavesdropping in the same laboratory.

## 3. Quantum Secret Sharing Scheme Based on *n* Particle GHZ State

### 3.1. Initial Stage

The measurement bases are $B_{x}=\{|x_{0}\rangle ,|x_{1}\rangle \}$ and $B_{y}=\{|y_{0}\rangle ,|y_{1}\rangle \}$, which can be expressed as follows:(1)$$\begin{array}{c} |x_{0}\rangle =\frac{1}{\sqrt{2}}(|0\rangle +\left|1\rangle ),\right|x_{1}\rangle =\frac{1}{\sqrt{2}}(|0\rangle -|1\rangle ),\\ |y_{0}\rangle =\frac{1}{\sqrt{2}}(|0\rangle +i|1\rangle ),|y_{1}\rangle =\frac{1}{\sqrt{2}}(|0\rangle -i|1\rangle ). \end{array}$$
By calculating, the basis |0〉,|1〉 can be expressed as
(2)$$|k\rangle =\frac{1}{\sqrt{2}}\sum_{l=0}^{1}e^{\pi kil}|x_{l}\rangle =\frac{1}{\sqrt{2}}\sum_{l=0}^{1}e^{\pi ki(l+\frac{3}{2})}|y_{l}\rangle ,$$
where k=0,1.

### 3.2. Distribution Stage

Participant $P_{A_{1}}$ randomly prepares a sequence of GHZ states consisting of $|GHZ_{i}\rangle_{1,\ldots ,n}$, where i∈{0,1}. Then, the first particle in each GHZ state is reserved, and the second to the p−th particle in each GHZ state are sent to $P_{A_{2}},\ldots ,P_{A_{p}}$, respectively. At the same time, the remaining *q* particles are sent to $P_{B_{1}},\ldots ,P_{B_{q}}$, respectively, as in the above. In this way, each participant obtained a sequence of particles. We use $L_{A_{1}},L_{A_{2}},\ldots ,L_{A_{p}}$ to represent these particle sequence of participants $P_{A_{1}},P_{A_{2}},\ldots ,P_{A_{p}}$, respectively, and $L_{B_{1}},L_{B_{2}},\ldots ,L_{B_{q}}$ to do the particle sequence of participants $P_{B_{1}},P_{B_{2}},\ldots ,P_{B_{q}}$.

### 3.3. Measurement Phase

After confirming that everyone has received these particles, $P_{A_{1}}$ randomly extracts some particles from his sequence $L_{A_{1}}$ as the detection particles, and informs others of the position of the detection particles (i.e., which particles in $L_{A_{1}}$ will be the detection particles). All participants in group $P_{A}$ use the base $B_{x}=\{|x_{0}\rangle ,|x_{1}\rangle \}$ for measurement, and $P_{A_{1}}$ designates group $P_{B}$ to use the base $B_{x}=\{|x_{0}\rangle ,|x_{1}\rangle \}$ or $B_{y}=\{|y_{0}\rangle ,|y_{1}\rangle \}$ for measurement. For the entangled states $|GHZ_{0}\rangle_{1,\ldots ,n}$ and $|GHZ_{1}\rangle_{1,\ldots ,n}$, the following four measurements may occur.

If group $P_{A}$ and $P_{B}$ use base $B_{x}$ and $B_{x}$ to measure $|GHZ_{0}\rangle_{1,\ldots ,n}$, respectively, then (3)$$|GHZ_{0}\rangle_{1,\ldots ,n}=2^{-\frac{n-1}{2}}\sum_{\begin{array}{c} l_{1},\ldots ,l_{p},l_{1}^{\prime },\ldots ,l_{q}^{\prime }=0\\ l_{1}+\ldots +l_{p}+l_{1}^{\prime }+\ldots +l_{q}^{\prime }\equiv 0\;\left(\mathrm{mod}\;2\right) \end{array}}^{1}|x_{l_{1}}\rangle \ldots |x_{l_{p}}\rangle |x_{l_{1}^{\prime }}\rangle \ldots |x_{l_{q}^{\prime }}\rangle .$$

If group $P_{A}$ and $P_{B}$ use base $B_{x}$ and $B_{y}$ to measure $|GHZ_{0}\rangle_{1,\ldots ,n}$, respectively, then (4)$$\begin{array}{c} |GHZ_{0}\rangle_{1,\ldots ,n}=\left\{\begin{array}{cc} 2^{-\frac{n-1}{2}}\sum_{\begin{array}{c} l_{1},\ldots ,l_{p},l_{1}^{\prime },\ldots ,l_{q}^{\prime }=0\\ l_{1}+\ldots +l_{p}+l_{1}^{\prime }+\ldots +l_{q}^{\prime }\equiv 0\;\left(\mathrm{mod}\;2\right) \end{array}}^{1}|x_{l_{1}}\rangle \ldots |x_{l_{p}}\rangle |y_{l_{1}^{\prime }}\rangle \ldots |y_{l_{q}^{\prime }}\rangle , & \mathrm{if}\;q=4t.\\ 2^{-\frac{n-1}{2}}\sum_{\begin{array}{c} l_{1},\ldots ,l_{p},l_{1}^{\prime },\ldots ,l_{q}^{\prime }=0\\ l_{1}+\ldots +l_{p}+l_{1}^{\prime }+\ldots +l_{q}^{\prime }\equiv 1\;\left(\mathrm{mod}\;2\right) \end{array}}^{1}|x_{l_{1}}\rangle \ldots |x_{l_{p}}\rangle |y_{l_{1}^{\prime }}\rangle \ldots |y_{l_{q}^{\prime }}\rangle , & \mathrm{if}\;q=4t+2. \end{array}\right. \end{array}$$
where t is an integer.

If group $P_{A}$ and $P_{B}$ use base $B_{x}$ and $B_{x}$ to measure $|GHZ_{1}\rangle_{1,\ldots ,n}$, respectively, then (5)$$|GHZ_{1}\rangle_{1,\ldots ,n}=2^{-\frac{n-1}{2}}\sum_{\begin{array}{c} l_{1},\ldots ,l_{p},l_{1}^{\prime },\ldots ,l_{q}^{\prime }=0\\ l_{1}+\ldots +l_{p}+l_{1}^{\prime }+\ldots +l_{q}^{\prime }\equiv 1\;\left(\mathrm{mod}\;2\right) \end{array}}^{1}|x_{l_{1}}\rangle \ldots |x_{l_{p}}\rangle |x_{l_{1}^{\prime }}\rangle \ldots |x_{l_{q}^{\prime }}\rangle .$$

If group $P_{A}$ and $P_{B}$ use base $B_{x}$ and $B_{y}$ to measure $|GHZ_{1}\rangle_{1,\ldots ,n}$, respectively, then (6)$$\begin{array}{c} |GHZ_{1}\rangle_{1,\ldots ,n}=\left\{\begin{array}{cc} 2^{-\frac{n-1}{2}}\sum_{\begin{array}{c} l_{1},\ldots ,l_{p},l_{1}^{\prime },\ldots ,l_{q}^{\prime }=0\\ l_{1}+\ldots +l_{p}+l_{1}^{\prime }+\ldots +l_{q}^{\prime }\equiv 1\;\left(\mathrm{mod}\;2\right) \end{array}}^{1}|x_{l_{1}}\rangle \ldots |x_{l_{p}}\rangle |y_{l_{1}^{\prime }}\rangle \ldots |y_{l_{q}^{\prime }}\rangle , & \mathrm{if}\;q=4t.\\ 2^{-\frac{n-1}{2}}\sum_{\begin{array}{c} l_{1},\ldots ,l_{p},l_{1}^{\prime },\ldots ,l_{q}^{\prime }=0\\ l_{1}+\ldots +l_{p}+l_{1}^{\prime }+\ldots +l_{q}^{\prime }\equiv 0\;\left(\mathrm{mod}\;2\right) \end{array}}^{1}|x_{l_{1}}\rangle \ldots |x_{l_{p}}\rangle |y_{l_{1}^{\prime }}\rangle \ldots |y_{l_{q}^{\prime }}\rangle , & \mathrm{if}\;q=4t+2. \end{array}\right. \end{array}$$
From the above four measurement results, it can be found that the detection results of *n* participants are correlated (Table A1 and Table A2 in Appendix A); that is, as long as the measurement results of any n−1 participants are confirmed, the measurement results of the last participant can be accurately judged without any operation and measurement.

### 3.4. Detection Stage

After the two groups $P_{A}$ and $P_{B}$ were measured according to the requirements, the group leader $P_{A_{1}}$ randomly asked the members of these two groups to make public the sequence of measurement results, but did not make public his own measurement results. Then, he checked whether the correlation was satisfied according to the published results from n−1 participants and his own measurement results. Next, $P_{A_{1}}$ compares the measurement results with the initial state. If the measurement result is different from the initial state, $P_{A_{1}}$ ask to stop this round and start a new round. Otherwise, it continues to execute.

### 3.5. Recovery Phase

The participants of groups $P_{A}$ and $P_{B}$, respectively, measure the remaining particles in their particle sequences, where the participants of group $P_{A}$ measure particles with the base $B_{x}=\{|x_{0}\rangle ,|x_{1}\rangle \}$, and the members of group $P_{B}$ measure particles with base $B_{x}=\{|x_{0}\rangle ,|x_{1}\rangle \}$ or $B_{y}=\{|y_{0}\rangle ,|y_{1}\rangle \}$. Then they encode these measurement results as binary numbers.The encoding method of participants from group $P_{A}$ is: the measurement result is $|x_{0}\rangle$, corresponding to binary number 0, and the measurement result is $|x_{1}\rangle$, corresponding to binary number 1. Thus, each participant in group $P_{A}$ receives a string of private key sequence $K_{A_{i}}\left(i=1,\ldots ,p\right)$. Since $P_{A_{1}}$ knows every GHZ state, he encodes the entangled state into a binary sequence $K_{a}$; that is, the entangled state $|GHZ_{0}\rangle_{1,\ldots ,n}$ corresponds to the binary number 0 and $|GHZ_{1}\rangle_{1,\ldots ,n}$ corresponds to the binary number 1. The coding method of participants from group $P_{B}$ is: the measurement result is $|x_{0}\rangle$ or $|y_{0}\rangle$, corresponding to binary number 0, and $|x_{1}\rangle$ or $|y_{1}\rangle$, corresponding to binary number 1. In this way, each participant $P_{B_{i}}$ receives a private key sequence $K_{B_{i}}\left(i=1,\ldots ,q\right)$. In addition, the members from group $P_{B}$ encode the used measurement basis into a sequence of binary key $K_{b}$. That is, the measurement basis $B_{x},B_{y}$ corresponds to the binary numbers 0,1, respectively. Next, let $K_{A}$ denote the keys from the group $P_{A}$, and $K_{B}$ the keys from the group $P_{B}$; then the final secret message *K* can be obtained in the following ways:

(1) When q=4t, $K_{A}$ and $K_{B}$ can be obtained by $K_{A}=\sum_{i=0}^{p}K_{A_{i}}+K_{a}$, $K_{B}=\sum_{i=0}^{q}K_{B_{i}}$, where $K_{A_{i}}$ and $K_{B_{i}}$ satisfy with $\sum_{i=0}^{p}K_{A_{i}}+K_{a}=\sum_{i=0}^{q}K_{B_{i}}$. Then the secret message $K=\sum_{i=0}^{p}K_{A_{i}}+K_{a}$, $K=\sum_{i=0}^{q}K_{B_{i}}$.

(2) When q=4t+2, $K_{A}$ and $K_{B}$ can be obtained by $K_{A}=\sum_{i=0}^{p}K_{A_{i}}+K_{a}$, $K_{B}=\sum_{i=0}^{q}K_{B_{i}}+K_{b}$, where $K_{A_{i}}$ and $K_{B_{i}}$ satisfy with $\sum_{i=0}^{p}K_{A_{i}}+K_{a}=\sum_{i=0}^{q}K_{B_{i}}+K_{b}$. Then the secret message $K=\sum_{i=0}^{p}K_{A_{i}}+K_{a}$, $K=\sum_{i=0}^{q}K_{B_{i}}+K_{b}$.

## 4. Safety Analysis

In this section, we analyze the security of the proposed scheme, and use MATLAB simulation analysis to show the relationship between the amount of information the adversary can obtain and the probability of being discovered.

### 4.1. Intercept-and-Resend Attack

We assume that the eavesdropper is Eve, she intercepts the particles sent by participant $P_{A_{1}}$. After measurement, she forges a particle sequence with the same measurement result and sends it to the other participant. Only the particles from group $P_{B}$ are transmitted through the quantum channel in the distribution stage. Eve can only intercept the particles from group $P_{B}$. However, the fake particle sequence of Eve has no correlation with the particles of $P_{A_{1}}$, which means that Eve may have been detected in the detection stage. If Eve chooses the correct measurement base and sends faked identical particles to group $P_{B}$ participants after measurement, the detection can be evaded; if the measurement basis used by Eve is different from that used by participants from group $P_{B}$, there is a 1/2 probability that Eve will not be detected according to the correlation. We consider the worst case here, i.e., if Eve intercepts all particles sent to group $P_{B}$ and chooses the correct measurement base, then forges the same particle as the measurement result and sends it to the participants from group $P_{B}$, the probability that Eve successfully evades detection and obtains GHZ information for a GHZ state is $\frac{1}{2}\cdot \frac{1}{2}\cdot \frac{1}{2}=\frac{1}{8}$. Let us say there are *w* GHZ states in total, and Eve has a ${\left(\frac{1}{8}\right)}^{w}$ probability of getting the secret message *K* without being discovered. When the number of GHZ states increases, that is, *w* increases, the probability of Eve being detected increases. However, for ordinary attackers, the probability of Eve successfully avoiding eavesdropping and obtaining secret messages is much less than ${\left(\frac{1}{8}\right)}^{w}$.

### 4.2. Entanglement Measurement Attack

In this protocol, the particle states during transmission are all in the maximum mixed state, that is, $\rho =Tr(|0\rangle \langle 0|+|1\rangle \langle 1|=\frac{I}{2})$, and Eve cannot distinguish them directly. Therefore, Eve chose to perform an eavesdropping operation to obtain more information about a GHZ state where she tries to entangle the additional particle with a particle in a GHZ state in the quantum channel, and measure that additional particle. According to Stinespring’s extension theorem [35], Eve’s eavesdropping operation may occur on a larger Hilbert space. Let the unitary operator $\hat{F}$ act on $|GHZ_{0}\rangle_{1,\ldots ,n}$ and the additional particle |χ〉; then we can obtain a complex system quantum state $|\phi^{\prime }\rangle$. That is, $|\phi^{\prime }\rangle =\hat{F}|GHZ_{0}\rangle_{1,\ldots ,n}|\chi \rangle =\sum_{k=0}^{1}{|k\rangle }_{1}\ldots {|k\rangle }_{n}⊗\eta_{k}$, where the dimension of the additional particle is not limited. Participants measure the quantum state $|GHZ_{0}\rangle_{1,\ldots ,n}$ in the recovery phase, and the composite system has the following cases:

(1) Groups $P_{A}$ and $P_{B}$ measure $|\phi^{\prime }\rangle$ with the basis $B_{x}$ and $B_{x}$, respectively; then the composite system space can be expressed as (7)$$\begin{array}{cc} |\phi^{\prime }\rangle & =2^{-\frac{n}{2}}\sum_{\begin{array}{c} l_{1},\ldots ,l_{n}=0\\ l_{1}+\ldots +l_{n}\equiv 0\;\left(\mathrm{mod}\;2\right) \end{array}}^{1}|x_{l_{1}}\rangle \ldots |x_{l_{n}}\rangle \left(\eta_{0}+\eta_{1}\right)\\ \; & +2^{-\frac{n}{2}}\sum_{\begin{array}{c} l_{1},\ldots ,l_{n}=0\\ l_{1}+\ldots +l_{n}\equiv 1\;\left(\mathrm{mod}\;2\right) \end{array}}^{1}e^{\pi i(l_{1}+\ldots +l_{n})}|x_{l_{1}}\rangle \ldots |x_{l_{n}}\rangle \left(\eta_{0}-\eta_{1}\right).\; \end{array}$$

(2) Groups $P_{A}$ and $P_{B}$ measure $|\phi^{\prime }\rangle$ with the basis $B_{x}$ and $B_{y}$, respectively; then the composite system space can be expressed as

(i) When q=4t, (8)$$\begin{array}{cc} |\phi^{\prime }\rangle & =2^{-\frac{n}{2}}\sum_{\begin{array}{c} l_{1},\ldots ,l_{p},l_{1}^{\prime },\ldots ,l_{q}^{\prime }=0\\ l_{1}+\ldots +l_{p}+l_{1}^{\prime }+\ldots +l_{q}^{\prime }\equiv 0\;\left(\mathrm{mod}\;2\right) \end{array}}^{1}|x_{l_{1}}\rangle \ldots |x_{l_{p}}\rangle |y_{l_{1}^{\prime }}\rangle \ldots |y_{l_{q}^{\prime }}\rangle \left(\eta_{0}+\eta_{1}\right)\\ \; & +2^{-\frac{n}{2}}\sum_{\begin{array}{c} l_{1},\ldots ,l_{p},l_{1}^{\prime },\ldots ,l_{q}^{\prime }=0\\ l_{1}+\ldots +l_{p}+l_{1}^{\prime }+\ldots +l_{q}^{\prime }\equiv 1\;\left(\mathrm{mod}\;2\right) \end{array}}^{1}|x_{l_{1}}\rangle \ldots |x_{l_{p}}\rangle |y_{l_{1}^{\prime }}\rangle \ldots |y_{l_{q}^{\prime }}\rangle \left(\eta_{0}-\eta_{1}\right). \end{array}$$

(ii) When q=4t+2, (9)$$\begin{array}{cc} |\phi^{\prime }\rangle & =2^{-\frac{n}{2}}\sum_{\begin{array}{c} l_{1},\ldots ,l_{p},l_{1}^{\prime },\ldots ,l_{q}^{\prime }=0\\ l_{1}+\ldots +l_{p}+l_{1}^{\prime }+\ldots +l_{q}^{\prime }\equiv 1\;\left(\mathrm{mod}\;2\right) \end{array}}^{1}|x_{l_{1}}\rangle \ldots |x_{l_{p}}\rangle |y_{l_{1}^{\prime }}\rangle \ldots |y_{l_{q}^{\prime }}\rangle \left(\eta_{0}+\eta_{1}\right)\\ \; & +2^{-\frac{n}{2}}\sum_{\begin{array}{c} l_{1},\ldots ,l_{p},l_{1}^{\prime },\ldots ,l_{q}^{\prime }=0\\ l_{1}+\ldots +l_{p}+l_{1}^{\prime }+\ldots +l_{q}^{\prime }\equiv 0\;\left(\mathrm{mod}\;2\right) \end{array}}^{1}|x_{l_{1}}\rangle \ldots |x_{l_{p}}\rangle |y_{l_{1}^{\prime }}\rangle \ldots |y_{l_{q}^{\prime }}\rangle \left(\eta_{0}-\eta_{1}\right). \end{array}$$

According to the above situation, if Eve’s actions did not trigger an error rate in the detection phase. The equation $\eta_{0}=\eta_{1}$ must be satisfied. Therefore, the above cases are denoted as (10)$$|\phi^{\prime }\rangle =2^{-\frac{n}{2}}\sum_{\begin{array}{c} l_{1},\ldots ,l_{n}=0\\ l_{1}+\ldots +l_{n}\equiv 0\;\left(\mathrm{mod}\;2\right) \end{array}}^{1}|x_{l_{1}}\rangle \ldots |x_{l_{n}}\rangle \left(\eta_{0}+\eta_{1}\right)$$ (11)$$\begin{array}{c} |\phi^{\prime }\rangle =\left\{\begin{array}{cc} 2^{-\frac{n}{2}}\sum_{\begin{array}{c} l_{1},\ldots ,l_{p},l_{1}^{\prime },\ldots ,l_{q}^{\prime }=0\\ l_{1}+\ldots +l_{p}+l_{1}^{\prime }+\ldots +l_{q}^{\prime }\equiv 0\;\left(\mathrm{mod}\;2\right) \end{array}}^{1}|x_{l_{1}}\rangle \ldots |x_{l_{p}}\rangle |y_{l_{1}^{\prime }}\rangle \ldots |y_{l_{q}^{\prime }}\rangle \left(\eta_{0}+\eta_{1}\right), & \mathrm{when}\;q=4t.\\ 2^{-\frac{n}{2}}\sum_{\begin{array}{c} l_{1},\ldots ,l_{p},l_{1}^{\prime },\ldots ,l_{q}^{\prime }=0\\ l_{1}+\ldots +l_{p}+l_{1}^{\prime }+\ldots +l_{q}^{\prime }\equiv 1\;\left(\mathrm{mod}\;2\right) \end{array}}^{1}|x_{l_{1}}\rangle \ldots |x_{l_{p}}\rangle |y_{l_{1}^{\prime }}\rangle \ldots |y_{l_{q}^{\prime }}\rangle \left(\eta_{0}+\eta_{1}\right), & \mathrm{when}\;q=4t+2. \end{array}\right. \end{array}$$

According to Equations (10) and (11), the composite quantum state of the additional particle and the GHZ states particle is always a product state without the error rate occurring. Therefore, the entanglement measurement attack is unsuccessful.

### 4.3. Analysis of Safety Simulation Model

From these two attacks, it can be seen that the error rate occurred with Eve is closely related to the probability that she can successfully evade detection. Next, let us analyze the relationship between them. When Eve wants to entangle the additional particles with the GHZ state in order to eavesdrop messages, the composite system state composed of Eve’s additional particles and GHZ is an entangled state $\phi_{A_{1}E}$. Let $I_{Eve}$ denote the amount of information that Eve can extract from the entangled state $\phi_{A_{1}E}$, and γ denote the error probability that occurred with Eve. According to ref. [36], γ and $I_{Eve}$ have the following relationship: (12)$$I_{Eve}\leq -\left(1-\gamma \right)log_{2}\left(1-\gamma \right)-\gamma log_{2}\left(\frac{\gamma }{3}\right).$$

If there are *L* GHZ states as the detection quantum states in the detection stage, the probability *f* of Eve being detected is $f=1-{\left(1-\gamma \right)}^{L}$.

From the above analysis, security model equations can be obtained: (13)$$\begin{array}{c} \left\{\begin{array}{c} I_{Eve}\leq -\left(1-\gamma \right)log_{2}\left(1-\gamma \right)-\gamma log_{2}\left(\frac{\gamma }{3}\right)\\ f=1-{\left(1-\gamma \right)}^{L} \end{array}\right. \end{array}$$

Considering the value of the number of detection particles L, and performing simulation analysis through MATLAB, Figure 1 can be obtained.

From the above analysis, it can be seen that both the amount of information acquired by Eve and the probability of Eve being detected increase with increases in the error probability γ. When the error probability is the same, the greater the number of detected particles, the higher the probability of Eve being detected. From Equation (Equation 13) and this above figure, when the error probability γ∈[0.739,0.761], the amount of information Eve can obtain reaches the maximum value 1, at which the information about the GHZ state can be obtained completely. The information obtained by Eve is $I_{Eve}\geq 1$, and the probability of Eve being detected is $f\geq \frac{1}{2}$. The number of detected particles L≥3.3029 can be obtained by solving. That is to say, as the number of detected particles *L* increases with L≥4, the probability of Eve being detected increases. Therefore, in the example of Section 6, we choose the number of detected particles L=4.

In addition, the probability of Eve being detected will increase as a convex function when the number of detected particles is small, then the probability of Eve being detected when acquiring information at the initial state is small and the security is low; in contrast, when the number of detected particles is large, the probability of Eve being detected will increase as a concave function. Although the amount of information obtained in the initial state is small, the probability of Eve being detected is very large, and the security of the scheme is increased. Therefore, when the number of detected particles is larger, the security of the protocol is higher. Let γ=0, that is, when the error that occurred with Eve is about 0, $I_{Eve}\approx 0$, this result is consistent with the result of entanglement measurement attack analysis.

## 5. Performance Comparison

In the following, we compare and analyze the literature [34,37,38,39] with our scheme from five aspects: using quantum states, space dimension, detecting quantum states, information efficiency and achievable threshold. The common point between these studies and our scheme is that they all use local discrimination to realize secret sharing. First, efficiency is an important criterion for judging an agreement. Cabello [40] defines a qubit usage efficiency formula $\eta =\frac{b_{s}}{q_{t}}$, where $q_{t}$ represents the total number of qubits transmitted in the quantum channel, and $b_{s}$ represents the total number of shared classical bits. According to the efficiency formula, our scheme will share *m* bits of classical information, and its efficiency is $\eta =\frac{m}{n(m+L)}$, where *L* represents the number of GHZ states as eavesdropping detection. As can be seen from Table 1, *l* GHZ states in the Rahaman scheme [37] are used to share *m* secret bits, since l−m GHZ states are used to check eavesdropping. Then the information efficiency of their scheme is $\frac{m}{nl}\left(l>m\right)$. If the number of eavesdropping particles is L=l−m, the information efficiency of their scheme is the same as ours. The Bai scheme [39] uses *m* GHZ states to share *m* secret bits, and *u* single photons are prepared for each particle sequence as detection particles. Then it uses n(m+u) photons for sharing *m* bit information among *n* participants. Therefore, the information efficiency of Bai’s scheme is $\frac{m}{n(m+u)}$. For the Yang scheme [38] and our scheme, *m* GHZ states are used to share *m* secret bits, and *L* GHZ states are applied to detect eavesdropping. Therefore, the information efficiency of both schemes is $\frac{m}{n(m+L)}$. The scheme of Li [34] uses 2m two-dimensional generalized Bell states to share *m* secret bits, and each participant prepares nu single photons as detection particles. Therefore, this scheme with *n* participants has $n^{2}u$ single photons for eavesdropping detection, so the information efficiency is $\frac{m}{n(2m+nu)}$. Compared with the scheme of Li [34], our scheme reduces the number of eavesdropping particles, and each GHZ state corresponds to sharing one bit of classical information, so the information efficiency is improved.

From the point of view of resources, although Rahaman’s scheme, Yang’s scheme, Bai’s scheme and our scheme all use the GHZ state with *n* particles as the transmission state, the particles of the GHZ state in our scheme and Rahaman’s scheme are taken from the two-dimensional space, while each particle of the GHZ state in Yang’s scheme and Bai’s scheme is taken from the high-dimensional space. Here we denote the dimension of the space as k(k≥3). By comparison of these two kinds of quantum states, obviously, the quantum state in our paper is easier to prepare and the cost will be lower. Li’s scheme uses the generalized Bell state as the transmission state. Each state contains two particles which also come from the high-dimensional space. Therefore, compared with Li’s paper, the particles required by Li are more difficult to prepare.

## 6. Example

This protocol is extended to the quantum secret sharing scheme of the n(n≥3) particle GHZ state. When n=3, the detailed protocol process is given in Song’s scheme [33]. In her protocol, Alice is equivalent to the leader $P_{A_{1}}$ of group $P_{A}$, Bob is equivalent to the leader $P_{B_{1}}$ of group $P_{B}$, and Charlie is equivalent to other members of group $P_{B}$. Alice prepares a list of GHZ entangled states arbitrarily, and keeps the first particle of each GHZ state, then sends the second particle and the third particle to Bob and Charlie, respectively. In the measurement phase, Alice measures the particle with $B_{x}$ base, Bob and Charlie measure with the same base $B_{x}$ or $B_{y}$, and encode the result into binary numbers according to the same method as our protocol. It is easy to verify that Bob and Charlie together can restore Alice’s secret. Since the simulation model analysis detects the number of eavesdropping quantum states L≥3.3029, Song’s scheme is safe. Therefore, this paper takes n=4 and L=4 as an example to implement the protocol in detail.

### 6.1. Protocol Process

#### 6.1.1. Initial Stage

There are four participants who are divided into two groups $P_{A}$ and $P_{B}$, with $P_{A}=\left\{P_{A_{1}},P_{A_{2}}\right\}$ and $P_{B}=\left\{P_{B_{1}},P_{B_{2}}\right\}$. The structure is shown in Figure 2. As shown in Figure 2, these two groups only have unique communication in the distribution stage, and the same secret is obtained through coding.

#### 6.1.2. Distribution Stage

Two groups of participants $P_{A_{1}}$ prepare a column of four particles GHZ. They are as follows:$$\begin{array}{c} |GHZ_{0}\rangle_{1,2,3,4},|GHZ_{1}\rangle_{1,2,3,4},|GHZ_{0}\rangle_{1,2,3,4},{|GHZ_{1}\rangle }_{1,2,3,4},\\ |GHZ_{0}\rangle_{1,2,3,4},|GHZ_{1}\rangle_{1,2,3,4},|GHZ_{0}\rangle_{1,2,3,4},{|GHZ_{1}\rangle }_{1,2,3,4}. \end{array}$$
Participant $P_{A_{1}}$ keeps the first particle in each GHZ state, and sends the second particle to $P_{A_{2}}$ of Group $P_{A}$, then sends the third and fourth particles to $P_{B_{1}}$ and $P_{B_{2}}$ of group $P_{B}$ through the quantum channel. Thus, each participant holds a sequence of particles. We use $L_{A_{1}},L_{A_{2}},L_{B_{1}},L_{B_{2}}$ to represent the particle groups of participants $P_{A_{1}},P_{A_{2}},P_{B_{1}},P_{B_{2}}$, respectively. The structure is shown as (I) in Figure 3.

#### 6.1.3. Measurement Phase

After confirming that the other three participants have received particles, participant $P_{A_{1}}$ selects the second, third, fifth and eighth particles as the detection particles, and informs the other three participants of the location of these detection particles, The process is shown as (II) in Figure 3. All participants in group $P_{A}$ are measured with base $B_{x}=\{|x_{0}\rangle ,|x_{1}\rangle \}$, while $P_{A_{1}}$ assigns the second and fifth particles in group $P_{B}$ to be measured with base $B_{y}=\{|y_{0}\rangle ,|y_{1}\rangle \}$, and the third and eighth particles to be measured with base $B_{x}=\{|x_{0}\rangle ,|x_{1}\rangle \}$. Then the measurement results of the extracted three GHZ quantum states may appear as the following: (14)$$\begin{array}{cc} |GHZ_{1}\rangle_{1,2,3,4} & =\frac{1}{2\sqrt{2}}(|x_{0}\rangle |x_{0}\rangle |y_{0}\rangle |y_{0}\rangle +|x_{0}\rangle |x_{0}\rangle |y_{1}\rangle |y_{1}\rangle +|x_{0}\rangle |x_{1}\rangle |y_{0}\rangle |y_{1}\rangle +|x_{1}\rangle |x_{0}\rangle |y_{0}\rangle |y_{1}\rangle \\ \; & +|x_{0}\rangle |x_{1}\rangle |y_{1}\rangle |y_{0}\rangle +|x_{1}\rangle |x_{0}\rangle |y_{1}\rangle |y_{0}\rangle +|x_{1}\rangle |x_{1}\rangle |y_{0}\rangle |y_{0}\rangle +|x_{1}\rangle |x_{1}\rangle |y_{1}\rangle |y_{1}\rangle ).\\ |GHZ_{0}\rangle_{1,2,3,4} & =\frac{1}{2\sqrt{2}}(|x_{0}\rangle |x_{0}\rangle |x_{0}\rangle |x_{0}\rangle +|x_{0}\rangle |x_{0}\rangle |x_{1}\rangle |x_{1}\rangle +|x_{0}\rangle |x_{1}\rangle |x_{0}\rangle |x_{1}\rangle +|x_{1}\rangle |x_{0}\rangle |x_{0}\rangle |x_{1}\rangle \\ \; & +|x_{0}\rangle |x_{1}\rangle |x_{1}\rangle |x_{0}\rangle +|x_{1}\rangle |x_{0}\rangle |x_{1}\rangle |x_{0}\rangle +|x_{1}\rangle |x_{1}\rangle |x_{0}\rangle |x_{0}\rangle +|x_{1}\rangle |x_{1}\rangle |x_{1}\rangle |x_{1}\rangle ).\\ |GHZ_{0}\rangle_{1,2,3,4} & =\frac{1}{2\sqrt{2}}(|x_{0}\rangle |x_{0}\rangle |y_{0}\rangle |y_{1}\rangle +|x_{0}\rangle |x_{0}\rangle |y_{1}\rangle |y_{0}\rangle +|x_{0}\rangle |x_{1}\rangle |y_{0}\rangle |y_{0}\rangle +|x_{0}\rangle |x_{1}\rangle |y_{1}\rangle |y_{1}\rangle \\ \; & +|x_{1}\rangle |x_{0}\rangle |y_{0}\rangle |y_{0}\rangle +|x_{1}\rangle |x_{0}\rangle |y_{1}\rangle |y_{1}\rangle +|x_{1}\rangle |x_{1}\rangle |y_{0}\rangle |y_{1}\rangle +|x_{1}\rangle |x_{1}\rangle |y_{1}\rangle |y_{0}\rangle ).\\ |GHZ_{1}\rangle_{1,2,3,4} & =\frac{1}{2\sqrt{2}}(|x_{0}\rangle |x_{0}\rangle |x_{0}\rangle |x_{1}\rangle +|x_{0}\rangle |x_{0}\rangle |x_{1}\rangle |x_{0}\rangle +|x_{0}\rangle |x_{1}\rangle |x_{0}\rangle |x_{0}\rangle +|x_{0}\rangle |x_{1}\rangle |x_{1}\rangle |x_{1}\rangle \\ \; & +|x_{1}\rangle |x_{0}\rangle |x_{0}\rangle |x_{0}\rangle +|x_{1}\rangle |x_{0}\rangle |x_{1}\rangle |x_{1}\rangle +|x_{1}\rangle |x_{1}\rangle |x_{0}\rangle |x_{1}\rangle +|x_{1}\rangle |x_{1}\rangle |x_{1}\rangle |x_{0}\rangle ). \end{array}$$ According to the above results, we can find that the measurement results of the four participants are correlated, The measurement results are shown in Figure 3III.

#### 6.1.4. Detection Stage

After the four participants measured completely, $P_{A_{1}}$ randomly asked the other three participants to make public the order of measurement results, but did not make public his own measurement results. $P_{A_{1}}$ tested whether the correlation between Table A1 and Table A2 (in Appendix A) was satisfied according to the public results of the other three participants and his own measurement results. Here, we calculate a threshold considering the error rate of the quantum channel during transmission. If the error rate of the detection result is higher than this threshold, the communication will be abandoned. Otherwise, the agreement continues.

#### 6.1.5. Recovery Phase

Members of two groups $P_{A}$ and $P_{B}$ measure the remaining four particles, respectively. Group $P_{A}$ uses the basis $B_{x}=\{|x_{0}\rangle ,|x_{1}\rangle \}$ for all measurements, and $P_{B_{1}}$ uses the basis $B_{x},B_{x},B_{y},B_{y}$ to measure four particles, respectively. Since the measurement results of each particle may have two results, it is advisable to assume that the measurement results of the four particles held by $P_{A_{1}},P_{A_{2}},P_{B_{1}},P_{B_{2}}$ are $\{|x_{0}\rangle ,|x_{0}\rangle ,|x_{1}\rangle ,|x_{1}\rangle \},\{|x_{1}\rangle ,|x_{1}\rangle ,|x_{0}\rangle ,|x_{1}\rangle \},\{|x_{0}\rangle ,|x_{0}\rangle ,|y_{0}\rangle ,|y_{0}\rangle \},\{|x_{1}\rangle ,|x_{0}\rangle ,|y_{1}\rangle ,|y_{1}\rangle \}$, respectively. Thus, the original four GHZ entangled states collapse into the following situations:(15)$$\begin{array}{cc} \; & |GHZ_{0}\rangle =\frac{1}{2\sqrt{2}}|x_{0}\rangle |x_{1}\rangle |x_{0}\rangle |x_{1}\rangle ,\\ \; & |GHZ_{1}\rangle =\frac{1}{2\sqrt{2}}|x_{0}\rangle |x_{1}\rangle |x_{0}\rangle |x_{0}\rangle ,\\ \; & |GHZ_{1}\rangle =\frac{1}{2\sqrt{2}}|x_{1}\rangle |x_{0}\rangle |y_{0}\rangle |y_{1}\rangle ,\\ \; & |GHZ_{0}\rangle =\frac{1}{2\sqrt{2}}|x_{1}\rangle |x_{1}\rangle |y_{0}\rangle |y_{1}\rangle . \end{array}$$
The two groups of participants code after each measurement:
$$\mathrm{group}P_{A}:K_{A_{1}}=\left\{0011\right\},K_{A_{2}}=\{0011\},K_{a}=\left\{0110\right\},$$
$$\mathrm{group}P_{B}:K_{B_{1}}=\left\{0000\right\},K_{B_{2}}=\left\{1011\right\},K_{b}=\{0011\}.$$
Thus, groups $P_{A}$ and $P_{B}$ recover the secret message $K_{A}=\left\{1000\right\}$ and $K_{B}=\left\{1000\right\}$, respectively. The result is shown as (IV) in Figure 3.

### 6.2. Efficiency Analysis

In the above example, the number *L* of detected quantum states is equal to 4 and the number *m* of the classical bits shared by four participants is equal to 4. Therefore, according to the efficiency formula $\eta =\frac{m}{n(m+L)}$, the efficiency of this example is $\frac{1}{8}$.

### 6.3. Security Simulation Model Analysis

From the above, the number L of detected quantum states is equal to 4. The result is substituted into equation system (13) to obtain the following equation system
(16)$$\begin{array}{c} \left\{\begin{array}{c} I_{Eve}\leq -\left(1-\gamma \right)log_{2}\left(1-\gamma \right)-\gamma log_{2}\left(\frac{\gamma }{3}\right)\\ f=1-{\left(1-\gamma \right)}^{4} \end{array}\right. \end{array}$$
Figure 4 can be obtained by simulation analysis with MATLAB.

As can be seen from Figure 4, when the probability of detection of the external attacker Eve is 0.5, the information she can obtain is less than 1. Under such circumstance, we believe that this scheme is safe enough.

## 7. Conclusions

In this paper, we propose a multi-party and efficient quantum secret sharing scheme based on the GHZ entangled state of *n* particles. The scheme realizes the secret sharing in small groups through local measurement. Different from classical secret sharing, the consumption of quantum resources is one of the important criteria for judging quantum secret sharing schemes, and the information efficiency of our scheme is $\frac{m}{n(m+L)}$. This protocol theoretically proposes an *n* particle GHZ entangled state and a multi-party and efficient quantum secret sharing scheme. According to the latest domestic research, the entangled state of up to 18 qubits can be realized at present. Therefore, it is also worth studying the implementation and optimization of this protocol in the actual environment by using the existing entangled states. 

## Figures and Tables

**Figure 1 entropy-24-01433-f001:**
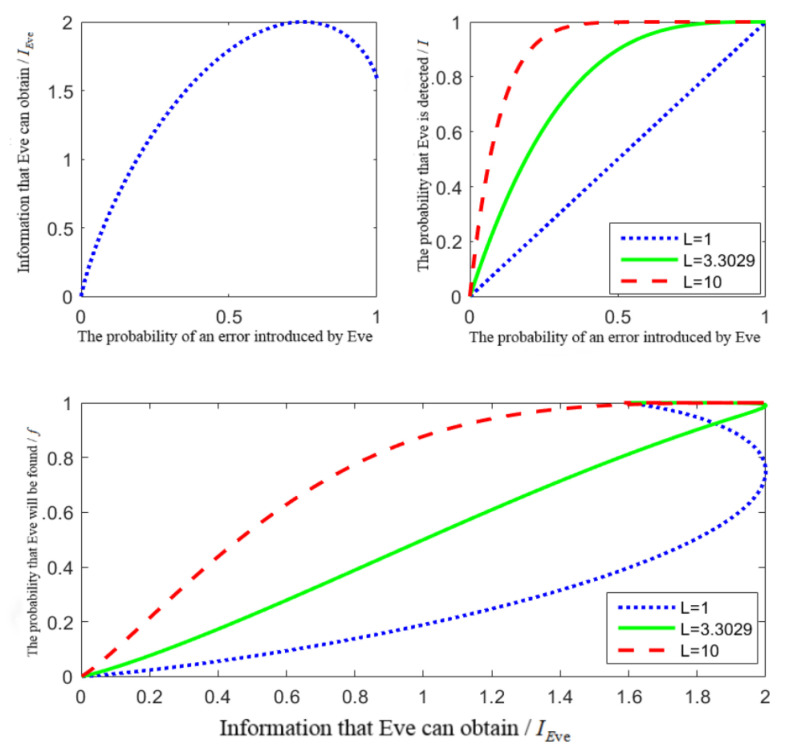
The relationship between $I_{Eve}$ and *f*.

**Figure 2 entropy-24-01433-f002:**
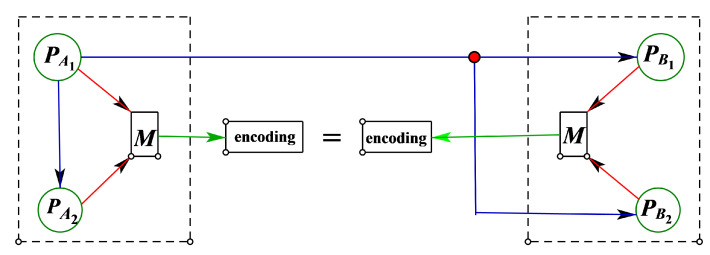
Square structure diagram.

**Figure 3 entropy-24-01433-f003:**
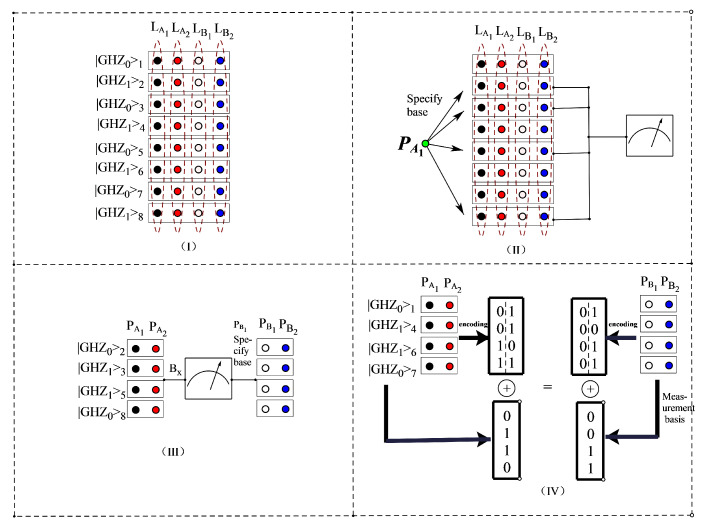
Flow chart of four particle GHZ states. ((**I**) represents the distribution stage, (**II**) represents the measurement phase, (**III**) represents the measurement results of the measurement phase, and (**IV**) represents the secret message that can finally be recovered.)

**Figure 4 entropy-24-01433-f004:**
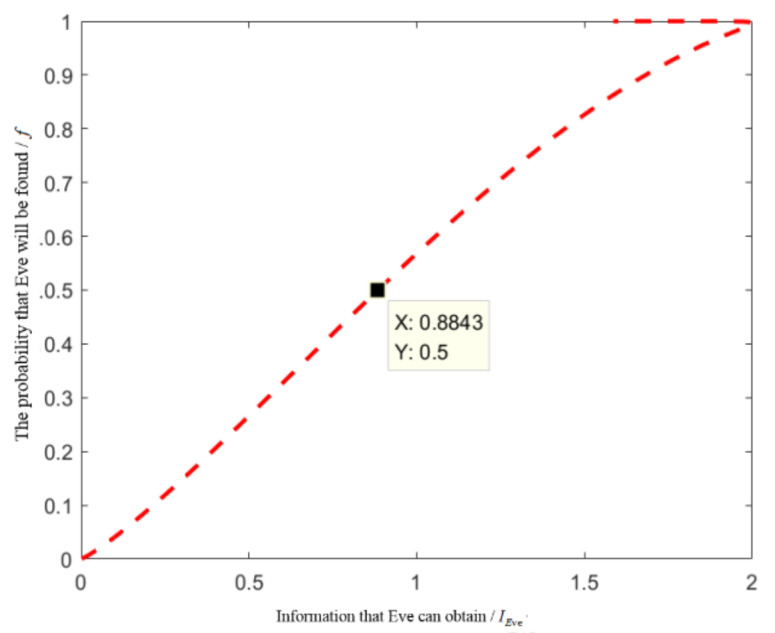
The relationship between $I_{Eve}$ and *f* when the quantum state L=4 is detected.

**Table 1 entropy-24-01433-t001:** Comparison between this scheme and existing schemes.

	Rahaman [37]	Yang [38]	Bai [39]	Li [34]	Our Scheme
Quantum states 1	GHZ state	GHZ state	GHZ state	The generalized Bell state	GHZ state
Dimension	2	k	k	k	2
Quantum states 2	GHZ state	GHZ state	single photon	single photon	GHZ state
Efficiency	$\frac{m}{n(m+L)}$	$\frac{m}{n(m+L)}$	$\frac{m}{n(m+u)}$	$\frac{m}{n(2m+nu)}$	$\frac{m}{n(m+L)}$
Threshold	R−(2,n)	(2,n)	R−(2,n)	(k,n)	R−(2,n)

In Table 1, Quantum states 1 denotes the used quantum states, and Quantum states 2 denotes the detected quantum states. *R* − (2, *n*) denotes the restricted QSS.

## Data Availability

The data on the correlation of the measured results in Section 3.3 (i.e., Table A1 and Table A2 in the Appendix A) are derived from reference [33,37].

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
