# Peer review of "Multi-Party Quantum Secret Sharing Based on GHZ State"

_entropy, 2022, doi:10.3390/e24101433_

Round 1
Reviewer 1 Report
The paper is generally well-written and can be of interest to the specialists in this area. I belive some minor adjustmenst can be done before the paper is accepted:
1. In Section 2.2 and the Conclusion the authors state that "If Eve obtains more than half of the information, and the probability of being detected is
greater than 1/2 at the same time, we consider that this scheme is safe." Since this is probably the most important parameter of the scheme, and it is not ideal, I advise the authors to justify why this metric is correct. If it's commonly accepted, they should provide references that support it. If different metrics are used, it should be explicitly mentioned, and they may be added to Table 1.
2. Section 6.1.4 mentiones Tables 2 and 3 but they can't be found in the paper
3. In line 170 it is unclear how to get 3/4 from the formula
4. In line 81 probably PBq should be in place of PBp
5. Eve is a female name, so she is normally referred as "She"
6. I believe "Intercept repeat" (sect. 4.1) is usually referred to as "Intercept-resend"
Reviewer 2 Report
The authors propose a scheme for secret sharing using GHZ state. They present the scheme and compare it to other approaches.
I think that the paper is interesting for people interested in the field but some clarifications are needed.
*) The formulas should be numbered.
*) In section 3.3 they write “and informs others of the position of the detection particle”. The authors should clarify what they mean by “position”.
*) In the final formula of section 3.3 I suppose that the state is GHZ_1 and not GHZ_0
After these points are addressed in my opinion the paper can be published.
